# Shape and Satellite Studies of Highly Charged Ions X-ray Spectra Using Bayesian Methods

Martino Trassinelli 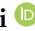

Institut des NanoSciences de Paris, CNRS, Sorbonne Université, 4 Place Jussieu, 75005 Paris, France; martino.trassinelli@insp.jussieu.fr

**Abstract:** High-accuracy spectroscopy commonly requires dedicated investigation into the choice of spectral line modelling to avoid the introduction of unwanted systematic errors. For such a kind of problem, the analysis of $\chi^2$ and likelihood are normally implemented to choose among models. However, these standard practices are affected by several problems and, in the first place, they are useless if there is no clear indication in favour of a specific model. Such issues are solved by Bayesian statistics, in the context of which a probability can be assigned to different hypotheses, i.e., models, from the analysis of the same set of data. Model probabilities are obtained from the integration of the likelihood function over the model parameter space with the evaluation of the so-called Bayesian evidence. Here, some practical applications are presented within the context of the analysis of recent high-accuracy X-ray spectroscopy data of highly charged uranium ion transitions. The method to determine the most plausible profile is discussed in detail. The study of the possible presence of satellite peaks is also presented.

**Keywords:** Bayesian methods; X-ray spectroscopy; model selection





## 1. Introduction

A problem commonly encountered in high-accuracy spectroscopy is the choice of the "right" profile to use for describing the different spectral components [1]. As an example, very recently it has been shown that the choice of the line profile (Gaussian or Voigt) is at the origin of the long-standing disagreement between theory and experiments on the oscillator-strength ratio in Fe XVII [2–4]. Another example is found in pionic atom spectroscopy, where the strong-interaction broadening can be extracted only by modelling a non-trivial Doppler broadening produced in the de-excitation cascade [5,6]. Standard methods to select the most adapted spectral line model are based on comparing the maximum value (or the $\chi^2$ minimum). Because of the consideration of maximum values only, for a chosen model these methods can however introduce systematic errors, such as biases, underestimation of uncertainties and artefacts in parameter correlations (see, e.g., [7,8]). This is not the case when the entire likelihood distribution is considered, as in Bayesian statistical methods. Moreover, standard methods provide only criteria and are useless if there is no clear indication in favour of a specific model. On the contrary, Bayesian methods, assigning a probability to models, allow us to deal with any possible scenario. A simple example of such problems is the choice of the best-adapted peak profile (Gaussian, Lorentzian, etc.) to model data such as those presented in Figure 1.

In the present article, the specific capabilities of Bayesian methods are illustrated in the analysis of high-accuracy X-ray spectroscopy of highly charged uranium ions. After a brief introduction on the model probability calculation (Section 2) and a short description of the data themselves (Section 3), we present the procedure to choose among different line models and to search for satellite lines (Section 4). Different strategies are considered and discussed. General considerations are presented in the last conclusive section (Section 5).

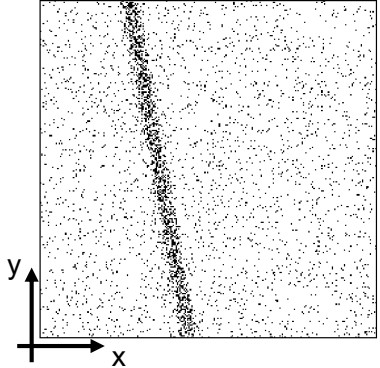 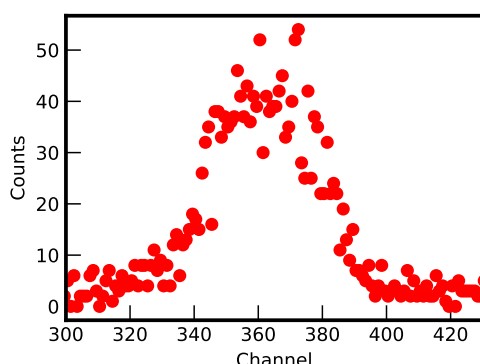

**Figure 1.** Spectral line corresponding to the He-like uranium $1s_{1/2}2p_{3/2} \rightarrow 1s_{1/2}2s_{1/2}$ intrashell transition obtained by a Bragg spectrometer. (**Left**): 2D spectrum from the spectrometer X-ray CCD with a binning of 8. The dispersion axis corresponds to the x-axis with $x$ proportional to the wavelength of the photon. (**Right**): Projection on the dispersion axis after line slope correction.

## 2. Bayesian Model Selection

In the context of Bayesian statistics, the probability of a model $\mathcal{M}$ for a given data set $\{x_i, y_i\}$ and some background information $I$ can be simply deduced by applying the Bayes' theorem [9–11] from the prior model probability $P(\mathcal{M}|I)$ and the Bayesian evidence $P(\{x_i, y_i\}|\mathcal{M}, I)$ of the model (also called marginal likelihood or model likelihood) with

$$P(\mathcal{M}|\{x_i, y_i\}, I) = \frac{P(\{x_i, y_i\}|\mathcal{M}, I) \, P(\mathcal{M}|I)}{P(\{x_i, y_i\}|I)}, \tag{1}$$

where $P(\{x_i, y_i\}|I)$ is a normalization factor.

The Bayesian evidence can be written as a function of the standard likelihood function $P(\{x_i, y_i\}|\boldsymbol{a}, \mathcal{M}, I)$ and the prior probability of the model parameters $P(\boldsymbol{a}|\mathcal{M}, I)$:

$$P(\{x_i, y_i\}|\mathcal{M}, I) = \int P(\{x_i, y_i\}|\boldsymbol{a}, \mathcal{M}, I)P(\boldsymbol{a}|\mathcal{M}, I)d^N\boldsymbol{a}, \tag{2}$$

where $\boldsymbol{a}$ are values of the $N$-dimensional parameters of the considered model.

Considering equal prior probabilities $P(\mathcal{M}|I)$ for the different models, the final probability $P(\mathcal{M}|\{x_i, y_i\}, I)$ is higher if the evidence is higher. A high value of evidence implies that the integral of the likelihood function over the entire parameter space is large, and not only its maximum, such as in methods based on the maximum likelihood. As a result of that, biases or other artefacts present in classical modelling are absent. Compared to function maximisation problems, the calculation of the integral in Equation (2) is most of the time not analytical and requires Monte Carlo sampling methods with large computation costs. Several approaches have been developed to reduce sampling time. Among them, the nested sampling method [12–14] is one of the most successful because of its special feature of reducing the multi-dimensional integral in Equation (2) to a one-dimensional integral and providing posterior probabilities of the inferred parameters as a by-product.

For cases with similar $P(\mathcal{M}|\{x_i, y_i\}, I)$, additional criteria can be used to select the most plausible model. One of these is the Bayesian complexity $\mathcal{C}$, which measures the number of model parameters that the data can support [10]. This quantity is related to the gain of information (in the Shannon sense) and it is calculated from the maximum value and average value of the likelihood function. Accordingly to Occam's razor, for similar $P(\mathcal{M}|\{x_i, y_i\}, I)$ the simplest model has to be selected, i.e., the model with the smallest $\mathcal{C}$. If independent datasets are available, additional indications can be obtained by the compatibility of the probability distributions of the model parameters from the different measurements.

## 3. Method

The data analysed in the present article are relative to the intrashell transitions $1s_{1/2}^p 2s_{1/2}^{q-1} 2p_{3/2}^1 \rightarrow 1s_{1/2}^p 2s_{1/2}^q$ from heliumlike (He-like, with $p = 1$, $q = 1$), lithium-like (Li-like, with $p = 2$, $q = 1$) and berylliumlike (Be-like, with $p = 2$, $q = 2$) uranium ions. Exited $1s_{1/2}^p 2s_{1/2}^{q-1} 2p_{3/2}^1$ levels of $U^{(92-p-q)+}$ ions are obtained by electron capture of stored $U^{(92-p-q+1)+}$ beams with a gaseous supersonic target and promptly de-excite via the emission of $\sim$4.5 keV photons. Such photons are detected by a pair of high-accuracy spectrometers based on Bragg diffraction from curved crystals placed at $\pm 90°$ observation angles, i.e., perpendicularly to the ion beam. The velocity of the ions is chosen to shift the photon energy to a common value around 4319 eV taking advantage of the relativistic Doppler effect. More information on the experimental setup can be found in a forthcoming paper [15] and in a similar previous experiment setup described in by the authors in [16].

Six spectra are considered in the present article—one per ion type per spectrometer (referred here as *inner* and *outer*, indicating their position with respect to the storage ring). Each spectrum is obtained by several hours of data acquisition on the X-ray CCDs of the two spectrometers resulting in images such as in Figure 1 (left), here relative to a He-like uranium $1s_{1/2}2p_{3/2} \rightarrow 1s_{1/2}2s_{1/2}$ intrashell transition. The corresponding slope correction and projection on the dispersion axis is presented in Figure 1 (right) . The characteristics of the spectral line depends on the natural width of the transition and the setup of the experiment. In addition to the general photon energy shift, the relativistic Doppler effect determines the spectral line slope due to the extension of the diffracting crystal, which corresponds to different observation angles around $\pm 90°$. Additionally, the line broadening along the dispersion axis is mainly caused by the Doppler effect and the size of the ion–target crossing region. This crossing region is determined by the intersection of the gaseous target, cylindrical with a uniform density, and the stored ion beam, with a Gaussian density distribution [17,18]. Because of the different ion velocities and beam preparations, line profiles from different ion species are expected to be different from each other, but similar between spectrometers. To simplify the analysis, the six spectra are considered to be independent of each other. Moreover, for the present article on the model selection of spectral lines, no particular prior assumption on the type of profile is made.

The 2D images are analysed using a recent version of the program `Nested_fit` (v4.0), which is based on the nested sampling algorithm [19–22]. Due to the low count rate per channel (with typical value of 0–2 in the spectral line), Poisson statistics are considered for the likelihood to avoid the introduction of possible biases [23,24]. Two-dimensional data are modelled with functions with a form

$$F[x, y] = f[x - x_0(y)] + C \quad \text{with} \quad x_0(y) = a + b(y - y_0), \tag{3}$$

where $y_0$ is the vertical coordinate of the middle of the detector and $C$ is a constant modelling of a flat background.

The evaluation of the Bayesian evidence was carried out with 2000 live points repeating the same evaluation eight times to estimate the evidence uncertainty. This uncertainty is due to the numerical accuracy of the nested sampling method to evaluate the integral in Equation (2). Uniform distributions with boundaries determined by the coarse characteristics of the spectra (e.g., the double of total number of counts for intensities, the total width of the 2D image for the line position, etc.) have been considered to be prior probabilities of the different model parameters. The choice of very large prior distribution results in the independency of such a choice of the final results [1,11].

## 4. Model Comparison Results

### 4.1. Profile Selection

Several different $f[x - x_0]$ profile models have been considered for describing the experimental peaks. Among them, the most probable profiles (higher evidence) are: Gaussian profile $f[x - x_0] = A \exp[-(x - x_0)^2/(2\sigma^2)]$, super-Gaussian profile $f[x - x_0] =$

$A \exp[-(x - x_0)^4/(4\sigma^4)]$ and the profile obtained by the convolution between a Gaussian with standard deviation $\sigma$ and a rectangular distribution of width $w$ resulting in

$$f[x - x_0] = A \frac{\mathrm{ERF}\left[\frac{w-x+x_0}{\sqrt{2}\sigma}\right] + \mathrm{ERF}\left[\frac{w+x-x_0}{\sqrt{2}\sigma}\right]}{4w},$$
(4)

where "ERF" indicates the error function. This profile will be referred to here as *ERFPEAK* (because of the error function presence in its analytic expression). Lorentzian and other profiles result in a negligible probability.

The different values of the logarithm of the evidence are presented in Figure 2 for the inner (top-left) and the outer spectrometer (top-right) and for the different intrashell transitions. The profile ERFPEAK has been taken as an arbitrary reference; only the evidence difference with respect to it is displayed. The super-Gaussian profile is favoured for most of the datasets, but not for the He-like and Li-like data of the inner spectrometer. For this reason, it has to be excluded in favour of the ERFPEAK profile which is the most or second most probable model in all datasets. Please note that a difference of 2 and 11 in the log value of the evidence corresponds to 2.5 and 5 standard deviations and a *p*-value of 1% and $4 \times 10^{-7}$ in favour of the model with higher evidence. (A correspondence between evidence differences, sigmas and *p*-values can be found in [25,26]). ERFPEAK is indeed the most likely profile a priori, which reflects the nature of the X-ray source obtained by the crossing of a Gaussian ion beam with a uniform cylindrical gas jet target [17,18]. As visible in Figure 2 (bottom), all the considered models provide consistent line positions with each other, within the statistical uncertainties. Please note that a more rigorous analysis would consist of considering spectra pair from the two spectrometers at the same time for each transition. Because of the linearity of the problem, adding (in log) the single demonstrates that a similar analysis can be obtained, where the spectral line widths are considered independent, which confirms our conclusions. The use of a common profile width parameter should, in principle, enhance even more the contrast between the different profile choices.

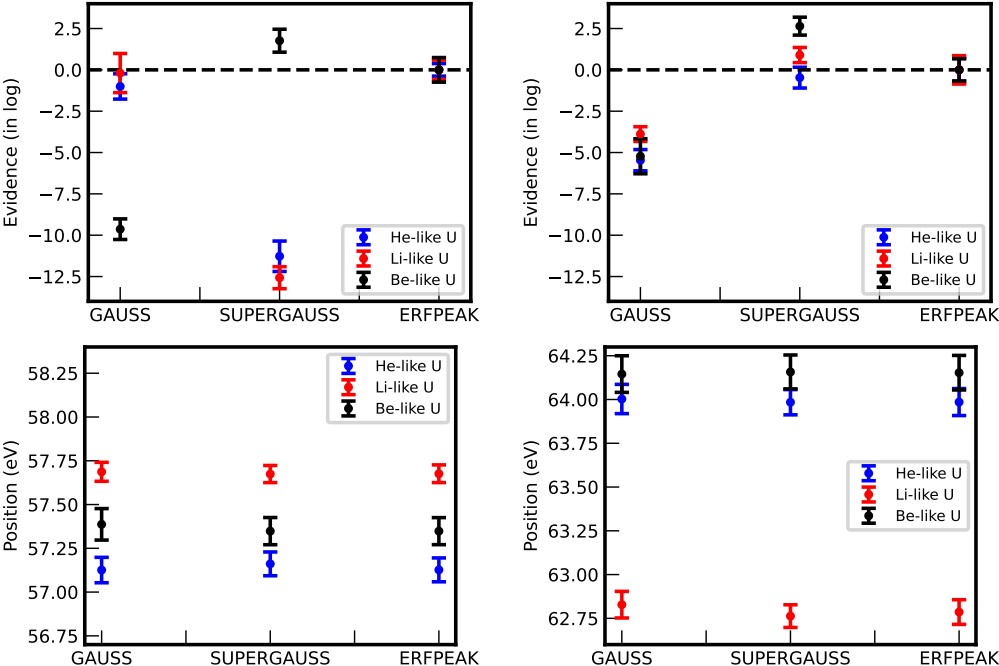

**Figure 2.** Plot of the relative evidence values (**top**) and spectral line position (**bottom**) for different line models for the spectra obtained by the inner (**left**) and outer (**right**) spectrometers.

*4.2. Search of Satellites*

Once the best adapted profile has been found, possible contamination from satellite contributions can be investigated. For this purpose, instead of one spectral line contribution, two close spectral lines with *ERFPEAK* profile with common width are considered. The component with the smaller amplitude is considered to be a satellite. For this purpose, the general model function

$$F[x,y] = f[x - x_0(y)] + \epsilon f[x - (x_0(y) + \Delta x)] + C \tag{5}$$

is considered, where $\epsilon$ and $\Delta x$ are relative amplitude and position of the satellite line with respect to the main component with uniform prior probabilities in the intervals $[0,1]$ and $[-70,70]$ channels, respectively. For this type of analysis, in addition to the Bayesian evidence and complexity, the final probability distributions of the satellite line parameters (Figure 3) are considered to check the consistency between pairs of spectra.

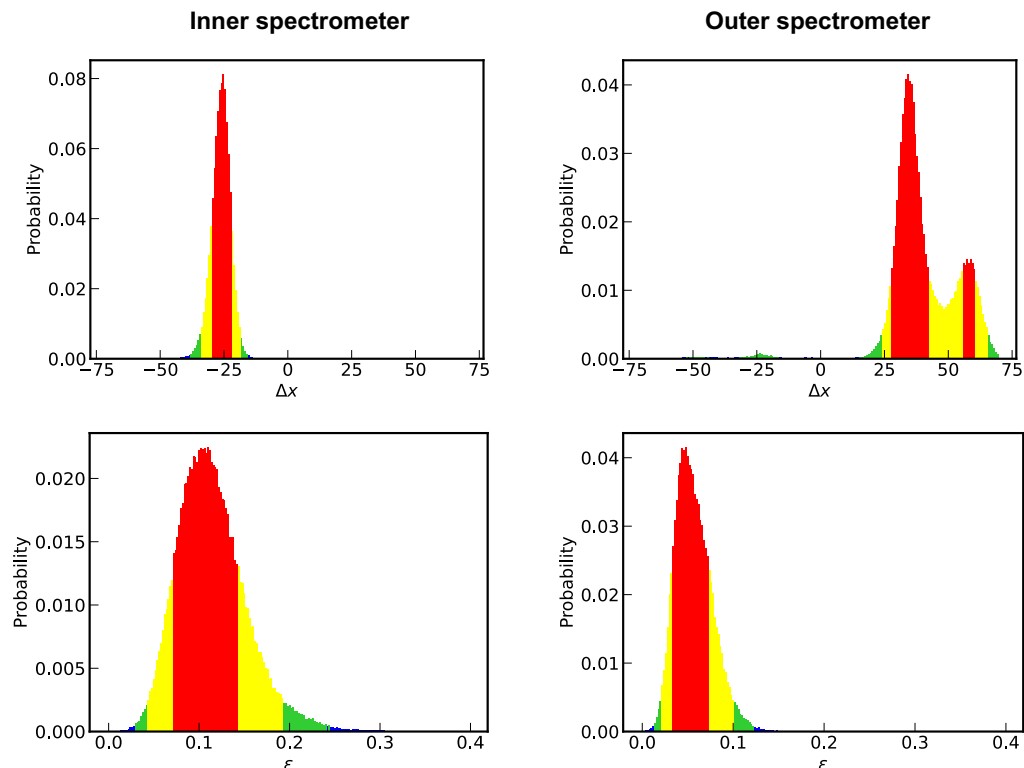

**Figure 3.** Probability distributions of the satellite line position (**top**) and amplitude (**bottom**) for the inner (**left**) and outer spectrometer (**right**) for the He-like line (Figure 1). Different colours correspond to one (red), two (yellow) and three (green) standard deviations intervals.

For Li-like and Be-like uranium transitions, one- and two-line models correspond to similar evidence values. The position of the main line does not significantly change, but the associated uncertainty increases (see Figure 4, left). For these cases, the Bayesian complexity $\mathcal{C}$ must be considered (see Section 2). For the considered transitions, models with satellite contribution present systematically higher complexity of about 8, compared to 6 when the satellite is not considered, which corresponds well to the number of parameters of the model. Consequently, the model with no satellite is the most plausible [10].

For He-like uranium, which is characterised by the lowest peak-to-background ratio, the Bayesian evidence evaluation considerably favours two-line models with a difference (in log) of slightly less than 5, which corresponds to 3.5 standard deviations and a *p*-value of $3 \times 10^{-4}$ (see Table 1). Thanks to the redundancy of the measurement with two independent spectra acquisitions from the two spectrometers, the posterior probability

of the position of the peak line in the two cases can be considered. As it can be observed in Table 1 and Figure 3, the satellite lines have not only different amplitudes but, more importantly, opposite positions with respect to the main line that are not compatible with each other (and where the increase of the x-axis corresponds in both cases to an increase of the wavelength of the detected X-ray). Because of that, even if the model with satellites is more probable for each spectrum, it must be discarded. This contradictory situation (high evidence but incompatible results) can be avoided with a more rigorous analysis as discussed in Section 4.1. Such an analysis consists of considering a likelihood function built from the two spectra of a same transition, each corresponding to one spectrometer, at the same time and modelling each spectrum with a pair of lines with the independent main line intensity and position and relative position and amplitude of the satellite line as common parameters.

**Table 1.** He-like uranium line data analysis selected outputs obtained from the evaluation of the Bayesian evidence of the spectra from the two spectrometers.

| | Evidence Difference | Equivalent $p$-Value | Mean Line Position Difference | Relative Satellite Position | Relative Satellite Amplitude |
|---|---|---|---|---|---|
| Inner arm | $4.8 \pm 1.1$ | $3.2 \times 10^{-4}$ | $0.29 \pm 0.12$ | $-25.8 \pm 3.9$ | $0.117 \pm 0.47$ |
| Outer arm | $4.9 \pm 1.2$ | $3.8 \times 10^{-4}$ | $-0.10 \pm 0.13$ | $40.1 \pm 14.7$ | $0.061 \pm 0.46$ |

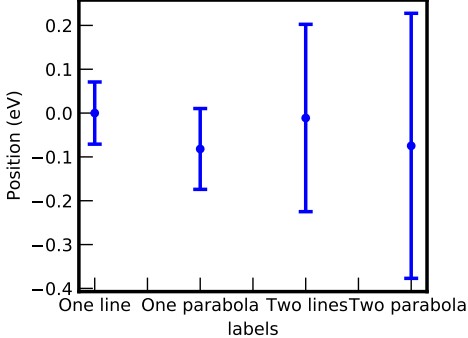 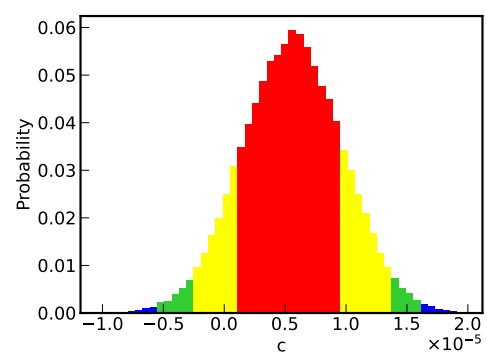

**Figure 4.** (**Left**): main spectral line position for different models. (**Right**): Probability distribution for the quadratic term of the spectral component for the Li-like line.

### 4.3. Spectral Line or Parabola

The almost point-like nature of the photon source and the spatial extension of the spectrometer position-sensitive detector generally determines the curved spectral [27,28]. However, if the source–detector path distance is large enough with respect to the detector extension, spectral lines can be considered to be straight lines. The use of lines or parabolas instead of straight lines can be directly investigated via Bayesian evidence evaluation. For this purpose, two choices of $x_0(y)$ are considered in $F[x, y]$ functions in Equations (3) and (5): $x_0(y) = a + b(y - y_0)$ or by $x_0(y) = a + b(y - y_0) + c(y - y_0)^2$. Li-like uranium spectra are exclusively considered for this type of analysis because their signal-to-background ratio is considerably higher than for the other spectra. Parabolic models systematically have a lower value of $-10$ than linear models (in log), which correspond to the very small $p$-value of $1 \times 10^{-6}$ and 4.8 standard deviations. Just as for the satellite lines, the position of the spectral component does not change significantly, but the associated uncertainty is deteriorated due to the higher number of model parameters (see Figure 4, left). Moreover, as shown in Figure 4 (right), the value of the additional parameter $c$ is compatible with a null value (within two standard deviations). This indicates the poor sensitivity to such a parameter with the present experimental data. For the above reasons,

a description with straight lines instead of parabolas is well justified, without the danger of introducing undesirable systematic effects.

## 5. Conclusions

In the previous sections, we presented a series of Bayesian methods to decide the best adapted profile in atomic spectra. Such methods are mainly based on the assignation of a probability to different models via the computation of Bayesian evidence. Different scenarios are encountered and discussed. The simplest case occurs when a model has a higher value of Bayesian evidence and, consequently, a higher probability. For models with similar probability, the Bayesian complexity must be taken into account, and the model with its lowest value should be considered. If independent measurements of the same atomic spectrum are available, a comparison of the final probability distribution of the parameter of the models is mandatory. An inconsistency in such distributions is a sign of incompatibility of the selected model for the considered data. The considerations discussed here for the specific case treated here can be simply implemented into any other spectra.

**Funding:** The results presented here are based on the experiment E125, which was performed at the infrastructure ESR at the GSI Helmholtzzentrum für Schwerionenforschung, Darmstadt (Germany) in the frame of FAIR Phase-0. This work has been partially supported by the European Union's Horizon 2020 research and innovation program and grant agreement n° 6544002 and the ExtreMe Matter Institute and Alexander von Humboldt Foundation.

**Conflicts of Interest:** The author declares no conflict of interest.

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
