# Peer review of "Shape and Satellite Studies of Highly Charged Ions X-ray Spectra Using Bayesian Methods"

_atoms, doi:10.3390/atoms11040064_

Round 1

Author Response

Please find the complete reply in the attached file.

Reviewer 2 Report

In this paper, the analysis method of line profiles and their parameters by Bayesian statistics is described using spectral data of He-like, Li-like, and Be-like uranium ions as examples. The validity of analytical results is confirmed by the fact that a profile with high probability agrees with that predicted from an experimental system. A method for finding the presence of satellite lines that partially overlap a main line is also mentioned. The reviewer finds the methodology of this paper interesting. However, the reviewer believes that the paper needs some modifications. The paper appears to consist in part of results and explanations of many of the analyses performed by the author that are consistent with the author's suggestions. The reviewer is encouraged to compare the results of the analysis and provide specific explanations in figures and tables so that the reader can objectively understand the appropriateness of this method. 

1) Specifics for Figures 1-4 and Table 1:

a) What are the experimental conditions and what line of what ion in Figure 1? 

b) In Figure 2, it is better to insert "inner" and "outer" in the figure to make it easier to understand.

c) Figures 3 and 4 lack explanation. In addition, the positioning of the figures and text appears to be poor.

d) Table 1 should clearly state in the caption or in the text how the results were analyzed.

2) The authors should write the definitions of Bayesian complexity C and p-value, which are mentioned in the paper.

3) The reviewer cannot imagine what 6 spectra analyzed by the author look like in ref.9 (in preparation) and ref.10. Is Figure 1 one of them?  The reviewer believes that one of the six within this paper would make it easier for the reader to understand.

4) Is the last paragraph of chapter 4 the topic of wavelength calibration (energy calibration) based on the analysis of line centers of the profiles?  It would be better if it is clear what the topic is.

Author Response

(The authors gave the same response as above.)
